# Canagliflozin Attenuates Lipotoxicity in Cardiomyocytes by Inhibiting Inflammation and Ferroptosis through Activating AMPK Pathway

**DOI:** 10.3390/ijms24010858

**Published:** 2023-01-03

**Authors:** Wanqiu Zhang, Jinghua Lu, Yangyang Wang, Pengbo Sun, Tong Gao, Naihan Xu, Yaou Zhang, Weidong Xie

**Affiliations:** 1State Key Laboratory of Chemical Oncogenomics, Shenzhen International Graduate School, Tsinghua University, Shenzhen 518055, China; 2Key Laboratory in Health Science and Technology, Institute of Biopharmaceutical and Health Engineering, Shenzhen International Graduate School, Tsinghua University, Shenzhen 518055, China; 3Open FIESTA Center, Shenzhen International Graduate School, Tsinghua University, Shenzhen 518055, China

**Keywords:** canagliflozin, lipotoxicity, cardiomyocytes, inflammation, ferroptosis, AMPK

## Abstract

Diabetic cardiomyopathy (DCM) is a myocardial disease independent of other cardiovascular diseases, such as coronary heart disease, hypertension, etc. Lipotoxicity is closely related to DCM. In this study, we investigated the mechanism of lipid metabolism disturbance in DCM in HL-1 cells. Through bioinformatics and Western blotting analysis, we found that canagliflozin (CAN) significantly inhibited the expression of inflammatory factors cyclooxygenase-2 (COX-2) and inducible nitric oxide synthase (iNOS). Ferroptosis is mediated by lipid peroxidation. We demonstrated the presence of ferroptosis in cardiomyocytes by detecting intracellular Fe^2+^ content and the levels of reactive oxygen species (ROS), malondialdehyde (MDA), reduced glutathione (GSH), and mitochondrial membrane potential (MMP). CAN could significantly regulate the indicators of ferroptosis. By using specific inhibitors celecoxib (coxib), S-methylisothiourea sulfate (SMT), Ferrostatin-1 (Fer-1), and Compound C, we further found that CAN regulated inflammation and ferroptosis through AMP-activated protein (AMPK), and inflammation interacted with ferroptosis. Our study indicated that CAN attenuated lipotoxicity in cardiomyocytes by regulating inflammation and ferroptosis through activating the AMPK pathway. This study provides a new direction of myocardial lipotoxicity and some new information for the treatment of DCM.

## 1. Introduction

According to the data of the International Diabetes Federation, the prevalence of diabetes among people aged 20–79 in the world is 10.5%, and China is the country with the largest number of diabetes patients [1]. Cardiovascular complications are a leading cause of death in diabetic patients. DCM was first proposed by Shirley Rubler in 1972, with altered myocardial structure and function independent of other cardiovascular diseases [2,3]. In diabetic patients, the main metabolic substrate switches from glucose to free fatty acid, a less efficient substrate than glucose in cardiac tissue, resulting in lower cardiac efficiency and further metabolic disorders [4]. When fatty acid levels exceed mitochondrial utilization capacity, it leads to the accumulation of lipids and toxic metabolites, resulting in mitochondrial dysfunction and cardiac lipotoxicity [5]. Lipotoxicity, directly impeding the metabolism of cardiomyocytes and promoting cells’ death [4], plays an important role in the development of DCM. Clinical trials indicated that CAN, a sodium glucose cotransporter 2 inhibitor, could significantly reduce the cardiovascular risks in diabetic patients [6,7,8]. Our previous study found that CAN improved heart failure caused by lipotoxicity and exerted anti-inflammatory effects [9,10,11]. It suggested that CAN may be a potential ideal drug for the treatment of DCM, but the specific mechanism needs further study.

Inflammation is a key pathogenic feature of DCM, which occurs in various diseases. The activation of NF-κB (p65) leads to leukocyte infiltration and upregulation of inflammatory pathways, resulting in severe myocardial damage [12]. Hyperglycemia, as well as increased free fatty acid metabolism, is associated with the upregulation of pro-inflammatory cytokines (e.g., IL-6, TNF-α) [12]. Inflammatory responses are involved in various processes, such as cardiomyocyte apoptosis, pro-fibrosis, and cardiomyocyte hypertrophy, impair cardiomyocyte contractility, and directly or indirectly promote the development of DCM [13]. Potential therapeutic strategies targeting inflammation are expected to show promising efficacy in future studies.

In 1981, Jerome L. Sullivan first proposed the hypothesis of “iron-derived heart disease”, but it was not until 2012 that Dixon discovered ferroptosis [14]. Ferroptosis might be an important breakthrough in the pathogenesis of heart disease, though there are few studies on the mechanism of ferroptosis in cardiomyocytes. The main mechanism of ferroptosis is to catalyze the lipid peroxidation of fatty acids on the cell membrane under the action of ferrous iron or ester oxygenase, thereby inducing cell death [15,16]. Iron accumulation, ROS, increased fatty acid supply, and lipid peroxidation are key to inducing ferroptosis [16]. Fang et al. observed the features of typical ferroptosis in doxorubicin-treated cardiomyocytes, revealing that ferroptosis might be a target for protection against cardiomyopathy [17].

We found that inflammatory factors were upregulated in palmitic acid (PA)-induced HL-1 cells. COX-2 and iNOS screened by gene chip were suggested to be an important cause of myocardial lipotoxicity, using molecular biological techniques. Ferroptosis, related to lipid peroxidation, also existed in PA-induced cardiomyocytes. By comparing CAN with coxib and SMT, we unexpectedly found that anti-inflammatory treatments inhibited ferroptosis. It suggested that CAN regulated ferroptosis, possibly by inhibiting inflammation. In addition, COX-2 and iNOS were inhibited by Fer-1. That is to say, inflammation and ferroptosis might interact with each other. We speculated that CAN attenuated lipotoxicity in cardiomyocytes, possibly by inhibiting inflammation and ferroptosis. Therefore, in this study, we investigated the mechanisms underlying the anti-lipotoxic effects of CAN on inflammation and ferroptosis.

## 2. Results

### 2.1. Canagliflozin Regulates Inflammatory Factors’ Expression in PA-Treated HL-1 Cells in Gene Chips

Up-regulated inflammatory pathways, leading to cardiomyocyte damage, directly or indirectly contribute to the development of DCM [13], but the key inflammatory mediators are unclear. We obtained some critical information of PA-induced HL-1 cells through gene sequencing, which included the normal control group, untreated PA control group, and CAN-treated PA group. Key inflammatory mediators were screened as potential targets for subsequent validation studies, using the data and online database. First, we obtained 456 genes highly expressed in gene chips by setting conditions (M_FPKM > 100). Next, we entered the keyword “inflammation” in the GeneCards database (The Human Gene Database, available online: https://www.genecards.org/, accessed on 17 May 2022) to obtain the top 200 inflammatory factors for subsequent analysis. The above data were further analyzed, and 13 core genes were screened using the Draw Venn Diagram software (Figure 1A and Appendix A). A heat map of 13 core genes was drawn according to the data in gene chips (Figure 1B and Appendix A). We found that some inflammatory genes were upregulated by PA and attenuated by CAN. It was reported that CAN had anti-inflammatory effects, but the mechanisms are still unclear [10]. After further analysis of the above results, we selected *Ptgs2* and *Nos2*, with higher expression and greater changes among the three groups, as potential targets for further studies. To confirm this, we tested the effect of CAN on the transcriptional levels of *Ptgs2*, *Nos2*, and the pro-inflammatory chemokines (e.g., *Ccl2*, *Ccl5*, and *Cxcl1*) and found that CAN significantly downregulated their mRNA levels, which were significantly upregulated by PA (Figure 1C–G).

### 2.2. Canagliflozin Inhibits the Expression of COX-2 and iNOS Proteins in PA-Treated HL-1 Cells

To further confirm the importance of inflammation, we tested the effect of CAN on the translation levels of COX-2 and iNOS in HL-1 cells. The cells were stimulated with 0.1 mM PA for 24 h, and the whole cell lysates were used for Western blotting analysis. We found that 5 μg/mL CAN showed a more significant and stable inhibition on protein levers of COX-2 and iNOS, which were significantly upregulated by PA (Figure 2A–C). Therefore, the dose of 5 μg/mL CAN was used for subsequent studies. COX-2 is a common inflammatory factor and may be a target for anti-inflammatory drugs [18]. Subsequently, we used coxib, a specific inhibitor of COX-2, for a comparative study. We found that CAN and coxib had comparable effects in regulating the protein levels of COX-2. Moreover, coxib showed better effect in inhibiting the protein levels of iNOS (Figure 2D–F). We speculated that COX-2 and iNOS interacted with each other. Therefore, we used SMT, an inhibitor of iNOS, to further investigate the interaction of COX-2 and iNOS. Compared with CAN and coxib, we found that SMT mainly affected the protein levels of iNOS and had little effect on COX-2 (Figure 2G–I). In addition, we found that CAN, coxib, and SMT all regulated the release of nitric oxide (NO), the downstream product of iNOS, and SMT showed a better inhibitory effect on the release of NO (Figure 2J,K).

### 2.3. Ferroptosis Is Observed in PA-Treated HL-1 Cells

Increased fatty acid β-oxidation and the imbalance of fatty acid uptake and oxidation cause myocardial lipotoxicity in DCM, but the specific mechanism is unclear [13]. Ferroptosis, a hot topic of recent research, was reported to exist in cardiomyocytes [17]. Therefore, we speculated that ferroptosis might play an important role in the development of DCM. To confirm this, we tested the key indicators of ferroptosis in PA-treated HL-1 cells, which mainly involved the redox state of cells. We found that Fer-1, the specific inhibitor of ferroptosis, significantly inhibited the death of cells, iron accumulation, increased MDA and ROS, decreased GSH, and increased MMP (Figure 3A–J and Appendix A).

### 2.4. Canagliflozin Regulates Ferroptosis Indexes in PA-Treated HL-1 Cells through Inflammatory Inhibition

Next, we evaluated the inhibitory effect of CAN on ferroptosis and conducted a comparative study using coxib and SMT, mentioned above. We unexpectedly found that CAN, coxib, and SMT all significantly inhibited the accumulation of iron, increased MDA and ROS, decreased GSH, and increased MMP and cell death in PA-treated HL-1 cells (Figure 4A–J and Appendix A). These results indicated that inflammation might affect ferroptosis and CAN might regulate ferroptosis by an anti-inflammatory mechanism.

### 2.5. Canagliflozin Regulates Ferroptosis-Mediated Inflammatory Damage

The above results showed that anti-inflammation could inhibit ferroptosis. Therefore, next, we studied the effect of ferroptosis on inflammation. We found that Fer-1, a specific inhibitor of ferroptosis, significantly inhibited the protein expression of COX-2 and iNOS (Figure 5A–C). Additionally, we surprisingly found CAN, coxib, SMT, and Fer-1 had a comparable effect in inhibiting inflammation, through the comparative experiments (Figure 5A–C). These results indicated that ferroptosis could affect inflammation and CAN might exert an anti-inflammatory effect by inhibiting ferroptosis.

### 2.6. Canagliflozin Promotes AMPK Activation through Upregulating LKB1, TAK1, and p-CaMKK2

Next, we investigated the mechanism of CAN inhibiting PA-induced inflammation and ferroptosis. Energy metabolism is abnormal in DCM. AMPK, as an important kinase regulating energy homeostasis, also plays a key role in DCM [3]. LKB1, TAK1, and CaMKK2 are key factors regulating AMPK. Through Western blotting analysis, we found that PA significantly downregulated the protein levels of LKB1 and TAK1 as well as the phosphorylation of CaMKK2 and AMPK, while CAN activated AMPK by upregulating LKB1, TAK1, and p-CaMKK2 (Figure 6A–G).

### 2.7. Canagliflozin Inhibits p-p65/COX-2/iNOS Inflammatory Pathway by Activating AMPK

AMPK regulates inflammation by inhibiting the NF-κB pathway [10]. CAN exerted an anti-inflammatory effect in immune cells by inhibiting the phosphorylation of p65 through activating AMPK [10]. In this study, we performed similar anti-inflammatory experiments and found that CAN significantly inhibited the phosphorylation of p65, which was upregulated by PA (Figure 7A,B). We used Compound C, a specific inhibitor of AMPK, to confirm the impact of AMPK on the protein levers of p-p65, COX-2, and iNOS and found that the inhibitory effect of CAN was indeed reversed (Figure 7C–G). These reversed effects of AMPK inhibition were also seen in coxib-treated cells. Therefore, we speculated that AMPK was an important target for inflammation regulation and CAN might inhibit the inflammatory pathway of p-p65/COX-2/iNOS by activating AMPK.

## 3. Discussion

Although it has been 50 years since DCM was first discovered, many mechanisms are still unclear. Lipotoxicity plays a key role in the development of DCM, so studying the mechanisms of lipotoxicity is crucial for the treatment of DCM. Currently, there is still no specific treatment for DCM, so “reuse of old drugs” is a fast, economical, and safe way to solve it, which has a significant advantage over developing new drugs. Previous studies have demonstrated that SGLT2 inhibitors could inhibit lipotoxicity and inflammation. CAN significantly reduced the levels of IL-1, IL-6, and TNF-α in cardiomyocytes [9]. This suggested that CAN alleviated lipotoxicity, possibly by reducing inflammation. Ferroptosis is closely related to lipid peroxidation [16]. Therefore, we hypothesized that there was also some link between ferroptosis and DCM. Hyperglycemia and disorders of lipid metabolism are the main features of DCM. In this study, we used a high-glucose medium to maintain cell growth and added PA, a widely used lipotoxicity modeling agent, to construct a lipotoxicity model in HL-1 cells. Sequentially, we investigated the protective mechanisms of CAN against lipotoxicity from both inflammation and ferroptosis. Our results showed that CAN significantly inhibited the expression of COX-2 and iNOS as well as the key indicators of ferroptosis, possibly due to the activation of AMPK, which in turn ameliorated inflammation and ferroptosis, alleviating lipotoxicity.

The anti-inflammatory effect of CAN has been reported in many studies. Niu et al. first reported that CAN attenuated NLRP3 inflammasome-mediated inflammation in immune cells by inhibiting NF-κB signaling and upregulating Bif-1, providing a new idea for the anti-inflammatory effect of CAN [10]. The discovery and selection of inflammatory targets might be accidental in many studies. In this study, we systematically screened the key inflammatory genes for further studies through bioinformatics analysis, which had better traceability and logic. Through subsequent repeated verification, we found that COX2 and iNOS screened from the gene chip might be important factors in mediating inflammation of cardiomyocytes. COX-2, an isoform of prostaglandin-endoperoxide synthase, acts in a pro-inflammatory manner, which is induced by multiple intracellular and extracellular stimuli [18]. The iNOS, a member of the nitric oxide synthase family, is only expressed in induced or stimulated cells, especially by pro-inflammatory cytokines or lipopolysaccharide [19]. The high expression of COX-2 and iNOS might be an important reason for the myocardial lipotoxicity. Targeting inflammation might be a reasonable strategy to attenuate lipotoxicity. In our study, we found that CAN had a significant regulatory effect on these inflammatory mediators. In addition, using specific inflammatory inhibitors and CAN, our study suggested that COX-2 had a greater effect on iNOS while, conversely, iNOS had a milder effect on COX-2. Therefore, we speculated that COX-2 might be an upstream protein of iNOS.

Several studies showed the existence of ferroptosis in cardiovascular diseases [17,20]. Ferroptosis, an iron-dependent, lipid peroxidation-driven programmed cell death, is different from apoptosis, autophagy, and pyroptosis [15]. Therefore, we suspected that targeting ferroptosis might be an effective strategy for protection against PA-treated HL-1 cells. Ferroptosis is mainly characterized by abnormal iron metabolism, accumulation ofROS, lipid peroxidation, glutathione metabolism disorders, and mitochondrial membrane potential (MMP) hyperpolarization [16,21]. As one of the key indicators, iron ion content can indicate whether ferroptosis occurs in cardiomyocytes [15]. An excess supply of fatty acids led to incomplete mitochondrial fatty acid oxidation, which caused an increase in redox pressure on the electron transport chain and ROS production, resulting in cellular oxidative stress and damage [22,23]. Lipid peroxidation plays a key role in cell damage and death in ferroptosis [16]. MDA, a peroxidative intermediate in the process of ferroptosis lipid peroxidation, is a key markers of lipid peroxidative damage. GSH, an antioxidant in animal cells, plays an important role in the antioxidant system of ferroptosis [15,20]. Studies showed that the MMP undergoes morphological changes of hyperpolarization in ferroptosis [21]. Therefore, based on the indicators, we evaluated whether ferroptosis was present in cardiomyocytes. Our results showed that Fer-1 corrected these indicators, indicating that ferroptosis did exist in cardiomyocytes. Recent studies showed that the SGLT2 inhibitor reduced ROS and had a significant inhibitory effect on MDA and SOD [9,22]. However, few studies systematically investigated the effect of CAN on ferroptosis. With this study, we were the first to discover that CAN could inhibit the indicators of ferroptosis in PA-treated HL-1 cells. Compared with specific inflammatory inhibitors, CAN was comparable in the regulation of ferroptosis. Anti-inflammation could inhibit the process of ferroptosis, which also suggested that CAN regulated ferroptosis, possibly mediated by inflammation inhibition. In addition, through assessing the effect of ferroptosis on COX-2 and iNOS, we found that ferroptosis mediates the occurrence of inflammation in PA-treated HL-1cells. That is, inflammation and ferroptosis interacted with each other. It suggested that CAN might ameliorate lipotoxicity by inhibiting inflammation and ferroptosis.

AMPK, known as a “metabolic master switch”, has many beneficial effects on cardio-metabolic abnormalities, such as myocardial inflammation, oxidative stress, and altered substrate utilization [3]. CAN exerts an anti-inflammatory effect by activating AMPK and inhibiting the NF-κB (p65) pathway [10]. P65 is essential for the expression of COX-2, by mediating the activity of a COX-2 promoter [24]. The iNOS induced by inflammatory signals is also regulated by p65 through directly binding to an *NOS2* promoter [19,25]. In addition, our studies above suggested that COX-2 might regulate the expression of iNOS. Therefore, we speculated that CAN inhibited inflammation in cardiomyocytes by inhibiting the inflammatory pathway of p-p65/COX-2/iNOS through activating AMPK. Our study demonstrated that the inactivation of AMPK indeed significantly weakened the inhibitory effect of CAN on inflammatory pathways, consistent with our hypotheses. AMPK also plays a role in the ferroptosis pathway. The biosynthesis of fatty acids is restrained and ferroptosis is inhibited when phosphorylated AMPK is activated by energy stress [26]. Collectively, our findings suggested that CAN alleviated lipotoxicity by inhibiting inflammation and ferroptosis, possibly due to the activation of AMPK.

Pro-inflammatory chemokines are a class of chemotactic cytokines that regulate cell migration and localization in inflammation [27]. In this study, we found that some chemokines were significantly upregulated in gene chips. Additionally, CAN had a significant inhibitory effect on the mRNA levers of *Ccl2*, *Ccl5*, and *Cxcl1* in PA-treated HL-1 cells. Previous studies discussed the importance of the chemokines above in acute inflammatory response and that *Ccl2* may be an effective target for the treatment of heart failure [27,28]. Therefore, we speculated that pro-inflammatory chemokines might play a role in the inflammation of DCM. In addition, our study only selected part of the results in the gene chips for detailed research; there are still many key genes that need to be further studied, such as *Serpine1*, *Vegfa*, etc. (Appendix A). A variety of bioinformatics technologies can be adopted to optimize the screening methods in the future, which will contribute to obtaining more accurate and comprehensive results through systematic research. Regarding changes of MMP in ferroptosis, MMP hyperpolarization and the loss of MMP were the main opinions [17]. We speculated that MMP was hyperpolarized in the early stage of ferroptosis, as we demonstrated before, and dissipated in the later stage, leading to mitochondrial dysfunction and severe cell damage. Our study demonstrated the existence of ferroptosis in PA-treated HL-1 cells, mainly through testing the key indicators of ferroptosis. Previous studies suggested that apoptosis was a pattern of death in cardiomyocytes [9], whereas we found that ferroptosis also occurred in cardiomyocytes. In addition, cell viability was only partially improved by reducing inflammation and ferroptosis in our study, probably because there are other ways for cell death, such as apoptosis, autophagy, pyroptosis, etc. Therefore, we speculated that there might be multiple patterns of death in cardiomyocytes. It still requires more data to further support our findings and hypotheses.

## 4. Materials and Methods

### 4.1. Cell Culture and Treatment

HL-1 cells (CL0683) were purchased from Fenghui Biotechnology Co., Ltd., Changsha, China. Cells were cultured in high-glucose DMEM (Gibco, Thermo Fisher Scientific, Waltham, MA, USA), supplemented with 10% fetal bovine serum (FBS; Gibco, Thermo Fisher Scientific, Waltham, MA, USA) and 1% penicillin-streptomycin antibiotic (Gibco, Thermo Fisher Scientific, Waltham, MA, USA) in an incubator of 5% CO_2_ at 37 °C. For inducing a lipotoxic cardiomyocyte model, cells were seeded into cell culture dishes or plates at an appropriate density and stimulated with 0.1 mM palmitic acid (PA; Sigma-Aldrich, Saint Louis, MO, USA) for 24 h in the presence or absence of CAN (5 μg/mL, BiochemPartner, Shanghai, China). Bovine serum albumin (BSA; 5%, g/mL, BIOFROXX, Einhausen, Germany) used to liquefy PA, was also added to blank control cells. COX-2 inhibitor celecoxib (5 μg/mL, TCI, Shanghai, China), iNOS inhibitor SMT (10 μM, Beyotime, Shanghai, China), and the ferroptosis inhibitor Fer-1 (5 μM, ABclonal Technology, Wuhan, China) were added simultaneously with PA. AMPK inhibitor Compound C (5 μg/mL, BiochemPartner, Shanghai, China) was added with PA and CAN.

### 4.2. Western Blotting Analysis

The whole cell lysates from HL-1 cells were collected, and the protein samples were separated using 10% sodium dodecyl sulfate polyacrylamide gel electrophoresis (SDS-PAGE; Epizyme Biotech, Shanghai, China). Then, they were transferred to nitrocellulose filter membranes (Pall, New York, NY, USA). Subsequently, the membranes were blocked with a fresh blocking buffer (5% g/mL nonfat milk powder (Epizyme Biotech, Shanghai, China), which was dissolved in tris-buffered saline with 0.2% Tween-20 (TBST)) for 2 h and then incubated with primary antibodies diluted with 1.5% BSA overnight at 4 °C. After a third rinse with TBST, the membranes were incubated with the secondary antibodies for 1 h at room temperature and then rinsed three more times with TBST. After that, the protein bands were visualized with a chemiluminescence solution (Thermo Fisher Scientific, Waltham, MA, USA). Finally, the ImageJ 1.48 software (National Institutes of Health, Bethesda, MD, USA) was used to quantify the relative gray density values of the protein bands.

The following were the specific primary antibodies used in the study: β-actin (1:5000, Mouse, A1978, Sigma-Aldrich, Saint Louis, MO, USA), COX-2 (1:1000, ET1610-23, HUABIO, Hangzhou, China), iNOS (1:1000, Rabbit, A3200, Abclonal Technology, Wuhan, China), LKB1 (1:1000, Rabbit, A2122, Abclonal Technology, Wuhan, China), TAK1 (1:1000, Rabbit, ET1705-14, HUABIO, Hangzhou, China), CaMKK2 (1:1000, Rabbit, A9899, Abclonal Technology, Wuhan, China), Phospho-CaMKK2 (Ser511) (1:1000, Rabbit, BD-PP1258, Biodragon, Suzhou, China), AMPKα (1:1000, Rabbit, 2532S, Cell Signaling Technology, Boston, MA, USA), Phospho-AMPKα (Thr172) (1:1000, Rabbit, 2535S, Cell Signaling Technology, Boston, MA, USA), NF-κB p65 (1:1000, Mouse, 6956S, Cell Signaling Technology, Boston, MA, USA), and Phospho-NF-κB p65 (Ser536) (1:1000, Rabbit, 3033S, Cell Signaling Technology, Boston, MA, USA). The following secondary antibodies were used: goat polyclonal antibody to rabbit IgG H&L HRP (1:5000, 7074P2, Cell Signaling Technology, Boston, MA, USA) and goat polyclonal antibody to mouse IgG H&L HRP (1:5000, 7076S, Cell Signaling Technology, Boston, MA, USA).

### 4.3. Quantitative Real-Time Polymerase Chain Reaction (RT-qPCR) Assay

Total RNA from HL-1 was extracted by AG RNAex Pro Reagent (Accurate Biology, Changsha, China) according to the manufacturer’s instructions. The concentration of the total RNA was measured by a NanoDrop 2000 spectrophotometer (Thermo Fisher Scientific, Waltham, MA, USA), and approximately 500 ng of total RNA was used for reverse transcription. Subsequently, cDNA was synthesized with Evo M-MLV RT Premix for qPCR (Accurate Biology, Changsha, China) and then quantitatively analyzed with SYBR Green Premix Pro Taq HS qPCR Kit (Accurate Biology, Changsha, China). Finally, the 2^−ΔΔCt^ method was used to calculate the relative gene expression.

The following were the sequences of primers used for RT-qPCR, which were synthesized by GENEWIZ, China: 5′-GGCTGTATTCCCCTCCATCG-3′ and 5′-CCAGTTGGTAACAATGCCATGT-3′ for *β-actin*, 5′-TGAGCAACTATTCCAAACCAGC-3′ and 5′-GCACGTAGTCTTCGATCACTATC-3′ for *Ptgs2*, 5′-GTTCTCAGCCCAACAATACAAGA-3′ and 5′-GTGGACGGGTCGATGTCAC-3′ for *Nos2*, 5′-GCTGCTTTGCCTACCTCTCC-3′ and 5′-TCGAGTGACAAACACGACTGC-3′ for *Ccl5*, 5′-TTAAAAACCTGGATCGGAACCAA-3′ and 5′-GCATTAGCTTCAGATTTACGGGT-3′ for *Ccl2*, and 5′-CTGGGATTCACCTCAAGAACATC-3′ and 5′-CAGGGTCAAGGCAAGCCTC-3′ for *Cxcl1*.

### 4.4. Thiazolyl Blue Tetrazolium Bromide (MTT) Assay

The cell viability was analyzed by MTT (Sangon Biotech, Shanghai, China) assay. Each well of cells was added 20 µL MTT (5 mg/mL) solution, and the cells were incubated for 2 h in an incubator of 5% CO_2_ at 37 °C. Then, we removed the cell medium and added 200 µL dimethyl sulfoxide (DMSO; Sangon Biotech, Shanghai, China) solution to thoroughly dissolve the purple formazan crystals. Optical density values at 490 nm (OD490) were analyzed by an Epoch microplate spectrophotometer (Bio-Tek, Winooski, VT, USA).

### 4.5. NO, MDA, and GSH Assays

Cell culture supernatants were collected and assessed by a Total Nitric Oxide Assay kit (S0023, Beyotime, Shanghai, China) for NO detection. The supernatants were collected after repeated freeze–thaw cycles of cells at −80 °C and assessed by GSH and GSSG Assay kit (S0053, Beyotime, Shanghai, China) for GSH detection. The whole cell lysates were collected and assessed by a Lipid Peroxidation MDA Assay Kit (S0131S, Beyotime, Shanghai, China) for MDA. They were all conducted according to the manufacturer’s instructions, and the protein concentration was determined by a Detergent Compatible Bradford Protein Assay Kit (P0006C, Beyotime, Shanghai, China).

### 4.6. Intracellular Ferrous Ion (Fe^2+^) Measurement

Intracellular Fe^2+^ was detected by a FeRhoNox-1 (MX4558, MKBio, Shanghai, China) fluorescent probe according to the manufacturer’s instructions. After one rinse with PBS buffer, the cells were incubated with 5 µM FeRhoNox-1 for 1 h in an incubator of 5% CO_2_ at 37 °C. Then, the fluorescence of Fe^2+^ under a Cy3 excitation filter was observed by a fluorescence microscope after two rinses with PBS buffer. Finally, ImageJ software was used to measure the fluorescence intensity of Fe^2+^.

### 4.7. ROS Assay

ROS was detected by a DCFH-DA (S0033S, Beyotime, Shanghai, China) fluorescent probe according to the manufacturer’s instructions. After one rinse with PBS buffer, the cells were incubated with 10 µM DCFH-DA (1:1000 diluted in serum-free culture medium) for 20 min at 37 °C. The fluorescence intensity was observed and measured after three rinses with a serum-free culture medium.

### 4.8. MMP Assay

MMP was detected by a mitochondrial membrane potential assay kit with JC-1 (C2006, Beyotime, Shanghai, China) according to the manufacturer’s instructions. The cells were collected and incubated with a JC-1 working solution for 25 min at 37 °C. After two rinses with JC-1 buffer (1×), the cells were resuspended with JC-1 buffer (1×) and filtered with 200-mesh cell sieves. Finally, the fluorescence signals were detected by flow cytometry and quantified by FlowJo_v10.8.1 software (BD, Ashland, OR, USA).

### 4.9. Statistical Analysis

All data were presented as the mean ± standard deviation (SD) in this study. Statistical differences among the different groups were analyzed by ANOVA followed by Tukey’s post hoc test. Statistical differences were considered significant at *p* < 0.05 and highly significant at *p* < 0.01.

## 5. Conclusions

In summary, we demonstrated that CAN significantly attenuated lipotoxicity in cardiomyocytes by inhibiting inflammation and ferroptosis through activating AMPK in vitro (Figure 8). Moreover, our study proposed some new perspectives to study myocardial lipotoxicity and reported the inhibitory effect of CAN on the p65/COX-2/iNOS signaling pathway and ferroptosis indicators, which may provide more data support for the application of CAN in cardiovascular diseases and new ideas for the treatment of patients with DCM. To further confirm the cardioprotective effects of CAN, additional related studies in vivo should be performed in the future.

## Figures and Tables

**Figure 1 ijms-24-00858-f001:**
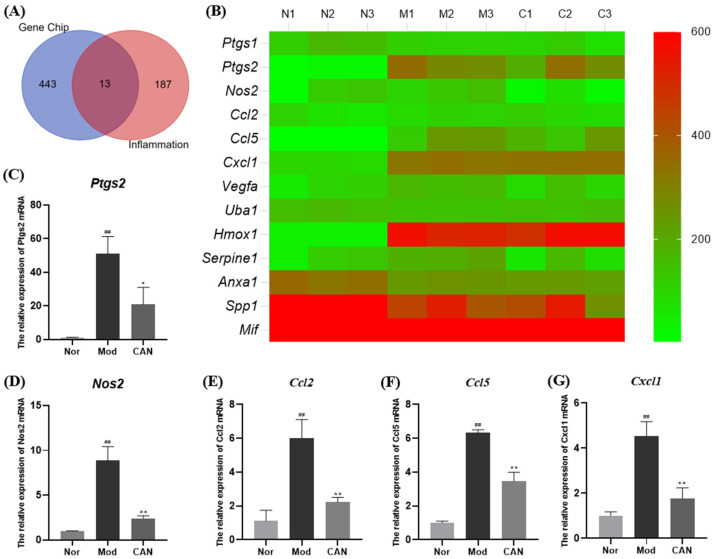
Canagliflozin regulates inflammatory factors’ expression in PA-treated HL-1 cells in gene chips. HL-1 cells were stimulated with 0.1 mM PA for 12 h in the presence or absence of CAN (5 μg/mL). (**A**) A total of 456 genes highly expressed in the gene chip (FPKM > 100) were intersected with the top 200 inflammatory factors in the GeneCards database by Draw Venn Diagram. Then, the Venn diagram of inflammatory genes in gene chips and GeneCards database was analyzed. (**B**) Heat map of 13 highly expressed inflammatory genes in gene chips. The mRNA levels of (**C**) *Ptgs2*, (**D**) *Nos2*, (**E**) *Ccl5*, (**F**) *Ccl2*, and (**G**) *Cxcl1* in HL-1 cells were quantified by RT-qPCR. Nor (N1, N2, N3), normal control group; Mod (M1, M2, M3), untreated PA control group; CAN (C1, C2, C3), CAN-treated PA group. Data were expressed as mean ± SD (*n* = 3). Differences were considered significant at *p* < 0.05 and highly significant at *p* < 0.01 (## *p* < 0.01 vs. Nor; * *p* < 0.05, ** *p* < 0.01 vs. Mod).

**Figure 2 ijms-24-00858-f002:**
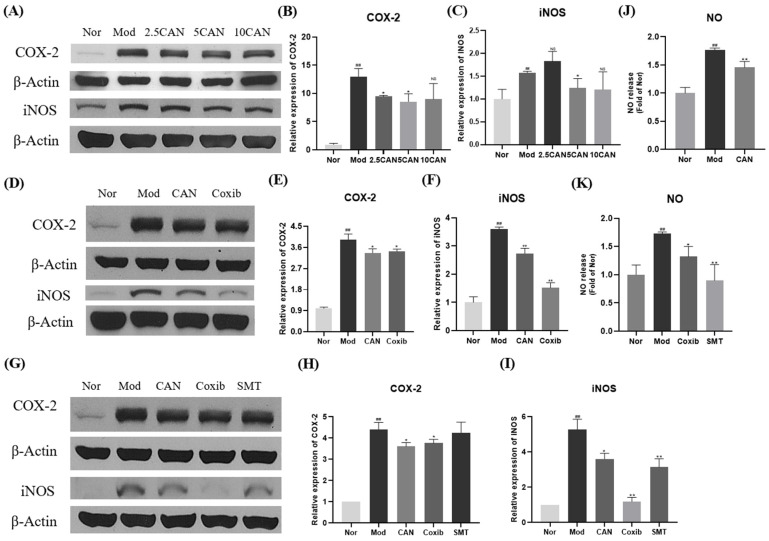
Canagliflozin inhibits the expression of COX-2 and iNOS proteins in PA-treated HL-1 cells. HL-1 cells were stimulated with 0.1 mM PA for 24 h, and the whole cell lysates from HL-1 cells were used for Western blotting analysis. (**A**–**C**) Effects of different doses of CAN on the levels of COX-2 and iNOS proteins in HL-1 cells. (**D**–**F**) Effects of COX-2 inhibitor coxib (5 μg/mL) on the levels of COX-2 and iNOS proteins in HL-1 cells. (**G**–**I**) Effects of iNOS inhibitor SMT (10 μM) on the levels of COX-2 and iNOS proteins in HL-1 cells. (**J**,**K**) Released NO was assessed by NO kit. Nor, normal control group; Mod, untreated PA control group; CAN, CAN-treated PA group; Coxib, coxib-treated PA group; SMT, SMT-treated PA group. Data were expressed as mean ± SD (*n* = 3). Differences were considered significant at *p* < 0.05 and highly significant at *p* < 0.01 (## *p* < 0.01 vs. Nor; * *p* < 0.05, ** *p* < 0.01 vs. Mod).

**Figure 3 ijms-24-00858-f003:**
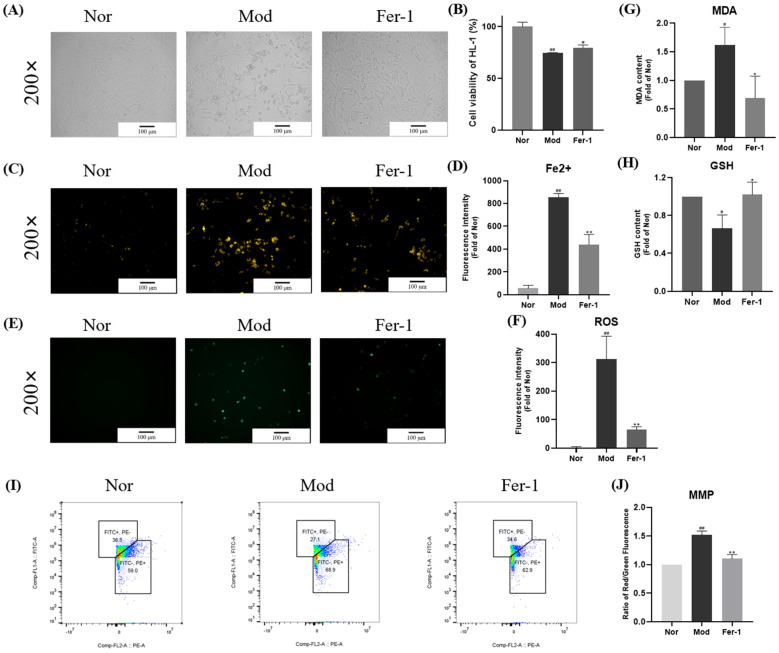
Ferroptosis is observed in PA-treated HL-1 cells. HL-1 cells were stimulated with 0.1 mM PA for 24 h in the presence or absence of Fer-1 (5 μM). (**A**) Representative images of live cells. (**B**) Effects of ferroptosis inhibitor Fer-1 on cell viability of HL-1 cells, which were analyzed by MTT assay. (**C**) Representative images of intracellular Fe^2+^ in HL-1 cells stained with FeRhoNox-1. (**D**) Quantification analysis of intracellular Fe^2+^. (**E**) Representative images of intracellular ROS in HL-1 cells stained with DCFH-DA. (**F**) Quantification analysis of intracellular ROS. (**G**) Intracellular MDA was assessed by MDA kit. (**H**) Intracellular GSH was assessed by GSH kit. (**I**,**J**) Quantification analysis of mitochondrial membrane potential in HL-1 cells by flow cytometry. Nor, normal control group; Mod, untreated PA control group; Fer-1, Fer-1-treated PA group. Data were expressed as mean ± SD (*n* = 3). Differences were considered significant at *p* < 0.05 and highly significant at *p* < 0.01 (# *p* < 0.05, ## *p* < 0.01 vs. Nor; * *p* < 0.05, ** *p* < 0.01 vs. Mod).

**Figure 4 ijms-24-00858-f004:**
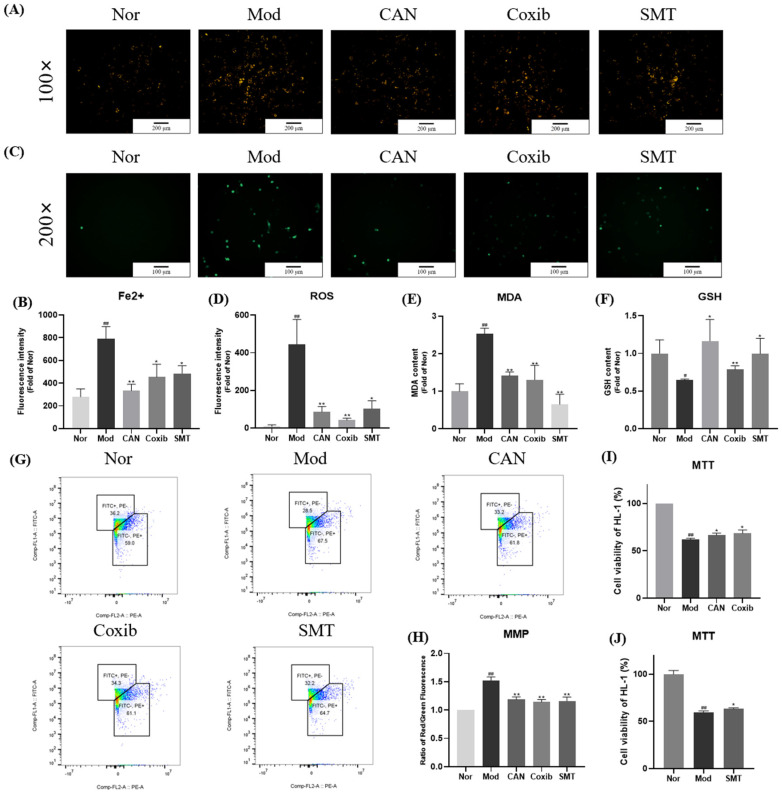
Canagliflozin regulates ferroptosis indexes in PA-treated HL-1 cells through inflammatory inhibition. HL-1 cells were stimulated with 0.1 mM PA for 24 h in the presence or absence of CAN (5 μg/mL), and the whole cell lysates from HL-1 cells were used for Western blotting analysis. Coxib (5 μg/mL) and SMT (10 μM) were used to explore the relationship between ferroptosis and inflammatory damage. (**A**) Representative images of intracellular Fe^2+^ in HL-1 cells stained with FeRhoNox-1. (**B**) Quantification analysis of intracellular Fe^2+^. (**C**) Representative images of intracellular ROS in HL-1 cells stained with DCFH-DA. (**D**) Quantification analysis of intracellular ROS. (**E**) Intracellular MDA was assessed by MDA kit. (**F**) Intracellular GSH was assessed by GSH kit. (**G**,**H**) Quantification analysis of mitochondrial membrane potential in HL-1 cells by flow cytometry. (**I**,**J**) Effects of CAN, coxib, and SMT on cell viability of HL-1 cells, which were analyzed by MTT assay. Nor, normal control group; Mod, untreated PA control group; CAN, CAN-treated PA group; Coxib, coxib-treated PA group; SMT, SMT-treated PA group. Data were expressed as mean ± SD (*n* = 3). Differences were considered significant at *p* < 0.05 and highly significant at *p* < 0.01 (# *p* < 0.05, ## *p* < 0.01 vs. Nor; * *p* < 0.05, ** *p* < 0.01 vs. Mod).

**Figure 5 ijms-24-00858-f005:**
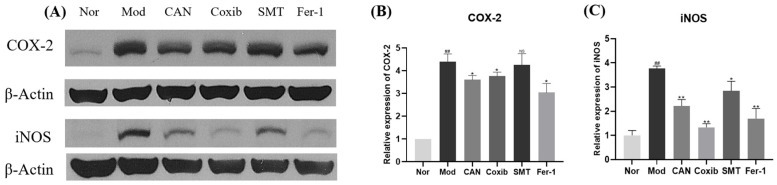
Canagliflozin regulates ferroptosis-mediated inflammatory damage. HL-1 cells were stimulated with 0.1 mM PA for 24 h in the presence or absence of CAN (5 μg/mL), and the whole cell lysates from HL-1 cells were used for Western blotting analysis. Coxib (5 μg/mL), SMT (10 μM), and Fer-1 (5 μM) were used to explore the relationship between ferroptosis and inflammatory damage. (**A**–**C**) Effects of CAN, coxib, SMT, and Fer-1 on the levels of COX-2 and iNOS proteins in HL-1 cells. Nor, normal control group; Mod, untreated PA control group; CAN, CAN-treated PA group; Coxib, coxib-treated PA group; SMT, SMT-treated PA group; Fer-1, Fer-1-treated PA group. Data were expressed as mean ± SD (*n* = 3). Differences were considered significant at *p* < 0.05 and highly significant at *p* < 0.01 (## *p* < 0.01 vs. Nor; * *p* < 0.05, ** *p* < 0.01 vs. Mod).

**Figure 6 ijms-24-00858-f006:**
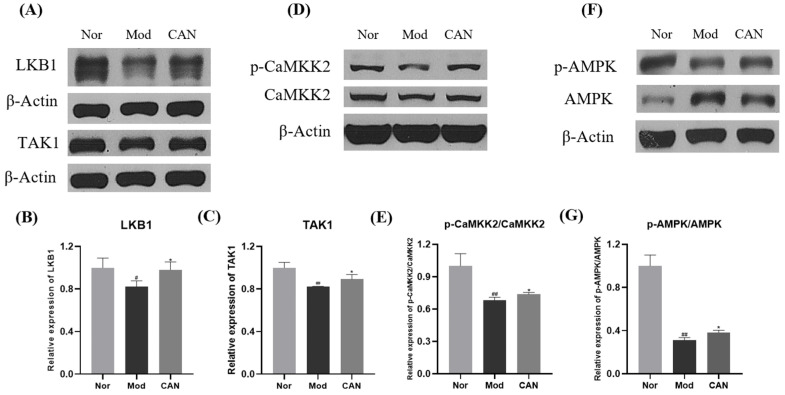
Canagliflozin promotes AMPK activation through upregulating LKB1, TAK1, and p-CaMKK2. HL-1 cells were stimulated with 0.1 mM PA for 24 h in the presence or absence of CAN (5 μg/mL), and the whole cell lysates from HL-1 cells were used for Western blotting analysis. (**A**–**C**) Effects of CAN on LKB1 and TAK1 in HL-1 cells. (**D**–**G**) Effects of CAN on the ratios of p-CaMKK2/CaMKK2 and p-AMPK/AMPK in HL-1 cells. Nor, normal control group; Mod, untreated PA control group; CAN, CAN-treated PA group. Data were expressed as mean ± SD (*n* = 3). Differences were considered significant at *p* < 0.05 and highly significant at *p* < 0.01 (# *p* < 0.05, ## *p* < 0.01 vs. Nor; * *p* < 0.05 vs. Mod).

**Figure 7 ijms-24-00858-f007:**
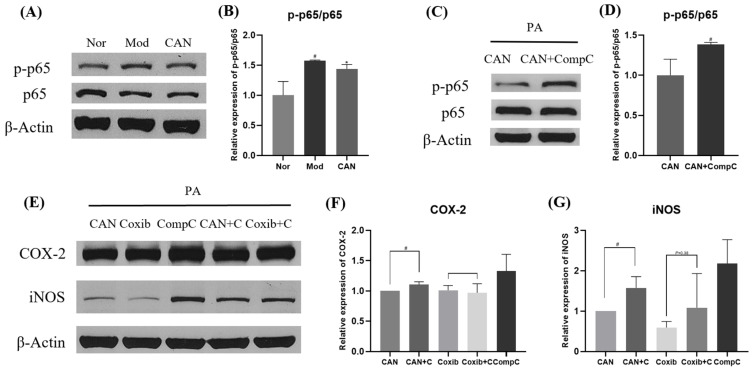
Canagliflozin inhibits p-p65/COX-2/iNOS inflammatory pathway by activating AMPK. HL-1 cells were stimulated with 0.1 mM PA for 24 h in the presence or absence of CAN (5 μg/mL), and the whole cell lysates from HL-1 cells were used for Western blotting analysis. (**A**,**B**) Effects of CAN on the ratios of p-p65/p65 in HL-1 cells. (**C**–**G**) HL-1 cells were treated with 0.1 mM PA and CAN (5 μg/mL)/Coxib (5 μg/mL) for 24 h in the presence or absence of Compound C (5 μg/mL), and the effects of AMPK inhibitor CompC on COX-2 and iNOS and the ratio of p-p65/p65 were analyzed. Nor, normal control group; Mod, untreated PA control group; CAN, CAN-treated PA group; Coxib, coxib-treated PA group; CompC, Compound C-treated PA group. Data were expressed as mean ± SD (*n* = 3). Differences were considered significant at *p* < 0.05 and highly significant at *p* < 0.01 (# *p* < 0.05 vs. Nor; * *p* < 0.05 vs. Mod).

**Figure 8 ijms-24-00858-f008:**
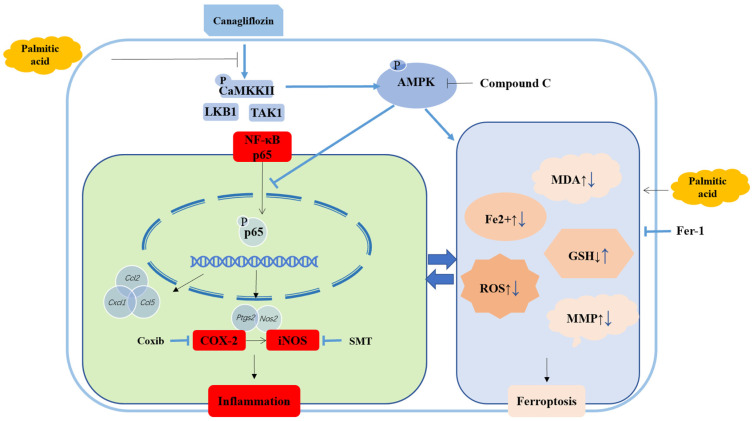
Schematic depicting that CAN attenuates lipotoxicity in cardiomyocytes by regulating p-p65/COX-2/iNOS-mediated inflammation and ferroptosis through activating AMPK pathway. Compound C, AMPK inhibitor; Coxib, COX-2 inhibitor; SMT, iNOS inhibitor; Fer-1, ferroptosis inhibitor.

## Data Availability

Not applicable.

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
