# Peer review of "Canagliflozin Attenuates Lipotoxicity in Cardiomyocytes by Inhibiting Inflammation and Ferroptosis through Activating AMPK Pathway"

_ijms, 2023, doi:10.3390/ijms24010858_

Round 1
Reviewer 1 Report
Study by Zhang at al. is really interesting, not only that it points out the need for investigation of potentially beneficial ("off") target effects of SGLT2 inhibitors, but also it brings feroptosis in the whole picture.
There are only couple of things I think might be added.
1. Authors mention gene array analysis, yet they only show data from the inflammatory pathway. As the PA treatment used seems to be very potent (40% reduction in viability) it would be nice to see the rest of highly expressed genes upon treatment as a proof of principle that system is indeed switching to beta oxidation. Also, would be cool to see what else is affected/rescued by CAN treatment (in combo with PA).
2. Along these lines, a condition with addition of insulin would have been beneficial to mimic hyperinsulinemia state, found in DCM. In this way, I think it hard to argue this a DCM model, rather only cardiomyocytes treated with PA. At least it would be good to elaborate a bit more why PA supplementation would be considered as DCM model.
Minor things:
- GSH, ROS, MDA and NO cannot have "expression". Authors should label the y-axes appropriately
- introduce PA abbreviation in the introduction
- remove the word "Authors " from the beginning of the discussion (line 245)
Author Response
Dear reviewer,
Thank you very much for your advice. We have revised the paper, and would like to resubmit it for your consideration. We have addressed the comments raised by you, and the revisions to the manuscript are marked up using the “Track Changes” function. Our responses to these comments, please see the attachment. We hope that the revision is acceptable, and we look forward to hearing from you soon.
Christmas is coming, have a great holiday!
With best wishes!
Yours sincerely,
Ms. Wanqiu Zhang
Shenzhen International Graduate School, Tsinghua University

Reviewer 2 Report
This study reports the mechanism of CAN to inhibit inflammation and ferroptosis in diabetic cardiomyopathy. It is well known that fatty acid could induce inflammation and ferroptosis, these are not novel. The interesting is this study shows the possibility to develop CAN for diabetic pateints who might get risk of cardiovascular diseases.
Before the acceptance for the publication in IJMS, there some points which must be clarified.
1. Figure 3b shows that PA slightly decreases cell survival and CAN exerts little effect to rescue the cells from lipotoxicity. These results may not represent the prevention ability of CAN inhibiting cell death induced by PA. Are there any other results which clearly shows PA dramatically stimulates cell death and CAN abrogates the effect.
2. Less effect of CAN is observed again in Figure 4I, the in vitro model used in this study may be not suitable to see CAN efficacy. Please dicuss about this.
3. Figure 7b is not related with figure7a.
4. Are there any other candidate molecules which are influence by CAN and involve in the inhibition of inflammation and cell death? Please discuss.
5. Please check your typo in the manuscript.
For example,
- western blot must be revised as Western blot
- line 28 cardomyocutes must be revised as cardiomyocytes
Author Response

(The authors gave the same response as above.)

Round 2
Reviewer 2 Report
This revised version could be accepted for the publication.